# DRUG DISCOVERY WITH DYNAMIC GOAL-AWARE FRAGMENTS

## ABSTRACT

Fragment-based drug discovery is an effective strategy for discovering drug candidates in the vast chemical space, and has been widely employed in molecular generative models. However, many existing fragment extraction methods in such models do not take the target chemical properties into account or rely on heuristic rules. Additionally, the existing fragment-based generative models cannot update the fragment vocabulary with goal-aware fragments newly discovered during the generation. To this end, we propose a molecular generative framework for drug discovery, named *Goal-aware fragment Extraction, Assembly, and Modification* (GEAM). GEAM consists of three modules, each responsible for goal-aware fragment extraction, fragment assembly, and fragment modification. The fragment extraction module identifies important fragments that contribute to the desired target properties with the information bottleneck principle, thereby constructing an effective goal-aware fragment vocabulary. Moreover, GEAM can explore beyond the initial vocabulary with the fragment modification module, and the exploration is further enhanced through the dynamic goal-aware vocabulary update. We experimentally demonstrate that GEAM effectively discovers drug candidates through the generative cycle of the three modules in various drug discovery tasks.

## 1 INTRODUCTION

The problem of drug discovery aims to find molecules with desired properties within the vast chemical space. Fragment-based drug discovery (FBDD) has been considered as an effective strategy in the recent decades as a means of exploring the chemical space and has led to the discovery of many potent compounds against various targets (Li, 2020). Inspired by the effectiveness of FBDD, many molecular generative models have also adopted it as a strategy to narrow down the search space and simplify the generation process, resulting in meaningful success (Jin et al., 2018; 2020a;b; Xie et al., 2020; Maziarz et al., 2022; Kong et al., 2022; Geng et al., 2023).

In FBDD, the first step, fragment library construction, directly impacts the final generation results (Shi & von Itzstein, 2019) as the constructed fragments are used in the entire generation process. However, existing fragment extraction or motif mining methods suffer from two limitations: they 1) do not take the target chemical properties of drug discovery problems into account and/or 2) rely on heuristic fragment selection rules. For example, it is a common strategy to randomly select fragments (Yang et al., 2021) or extract fragments based on frequency (Kong et al., 2022; Geng et al., 2023) without considering the target properties. Jin et al. (2020b) proposed to find molecular substructures that satisfy the given properties, but the extraction process is computationally very expensive and the substructures cannot be assembled together.

To this end, we first propose a novel deep learning-based goal-aware fragment extraction method, namely, *Fragment-wise Graph Information Bottleneck* (FGIB, Figure 1(a)). There is a strong connection between molecular structures and their activity, which is referred to as structure-activity relationship (SAR) (Crum-Brown & Fraser, 1865; Bohacek et al., 1996). Inspired by SAR, FGIB utilizes the graph information bottleneck (GIB) theory to identify important subgraphs in the given molecular graphs for predicting the target chemical property. These identified subgraphs then serve as building blocks in the subsequent generation. As shown in Figure 1(b), the proposed usage of goal-aware fragments extracted by FGIB improves the optimization performance by a significant margin compared to existing FBDD methods.

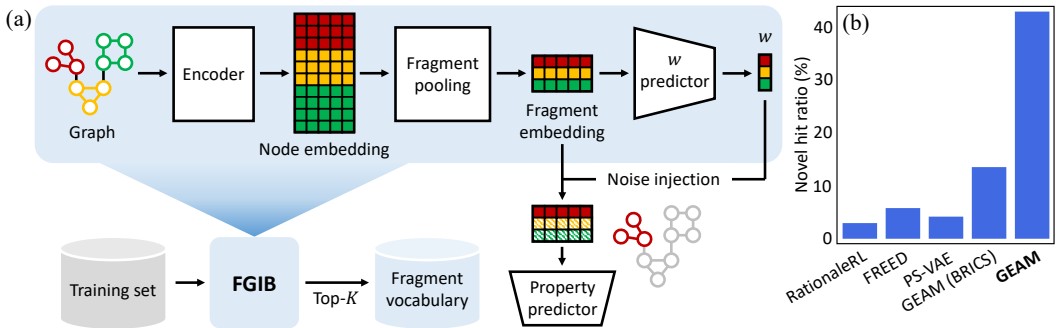

Figure 1: (a) **The architecture of FGIB.** Using the GIB theory, FGIB aims to identify the important subgraphs that contribute much to the target chemical property in the given molecular graphs. The trained FGIB is then used to extract fragments in a molecular dataset in the goal-aware manner. (b) **Performance comparison of GEAM and other FBDD methods** on the jak2 ligand generation task.

To effectively utilize the extracted fragments in molecular generation, we next construct a generative model consisting of a fragment assembly module and a fragment modification module. In this work, we employ soft-actor critic (SAC) for the assembly module and a genetic algorithm (GA) for the modification module. Through the interplay of the two modules, the generative model can both exploit the extracted goal-aware fragments and explore beyond the initial fragment vocabulary. Moreover, to further enhance molecular novelty and diversity, we propose to extract new fragments on-the-fly during the generation using FGIB and dynamically update the fragment vocabulary.

Taken as a whole, the fragment extraction module, the fragment assembly module, and the fragment modification module in the form of FGIB, SAC, and GA, respectively, collectively constitute the generative framework which we refer to as *Goal-aware fragment Extraction, Assembly, and Modification* (GEAM). As illustrated in Figure 2, GEAM generates molecules through the iterative process that sequentially runs each module as follows: 1) After FGIB constructs an initial goal-aware fragment vocabulary, SAC assembles these fragments and generates a new molecule. 2) GEAM keeps track of the top generated molecules as the initial population of GA, and GA generates an offspring molecule from the population. 3) As a consequence of the crossover and mutation procedures, the offspring molecule contains new subgraphs that cannot be constructed from the current fragment vocabulary, and FGIB extracts the meaningful subgraphs from the offspring molecule and update the vocabulary. Through the collaboration of the three modules where FGIB provides goal-aware fragments to SAC, SAC provides high-quality population to GA, and GA provides novel fragments to FGIB, GEAM effectively explores the chemical space to discover novel drug candidates.

We experimentally validate the proposed GEAM on various molecular optimization tasks that simulate real-world drug discovery scenarios. The experimental results show that GEAM significantly outperforms existing state-of-the-art methods, demonstrating its effectiveness in addressing real-world drug discovery problems. We summarize our contributions as follows:

- We propose FGIB, a novel goal-aware fragment extraction method that applies the GIB theory to construct a fragment vocabulary for target chemical properties.

- We propose to leverage SAC and GA jointly as a generative model to effectively utilize the extracted fragments while enabling exploration beyond the vocabulary.

- We propose GEAM, a generative framework that combines FGIB, SAC, and GA to dynamically update the fragment vocabulary by extracting goal-aware fragments on-the-fly to further improve diversity and novelty.

- We experimentally demonstrate that GEAM is highly effective in discovering drug candidates, outperforming existing molecular optimization methods.

## 2    RELATED WORK

**Fragment extraction**    Fragment extraction methods fragmentize the given molecules into molecular substructures, i.e., fragments, for subsequent generation. Yang et al. (2021) chose to randomly select fragments after breaking bonds in the given molecules with a predefined rule. Xie et al. (2020)

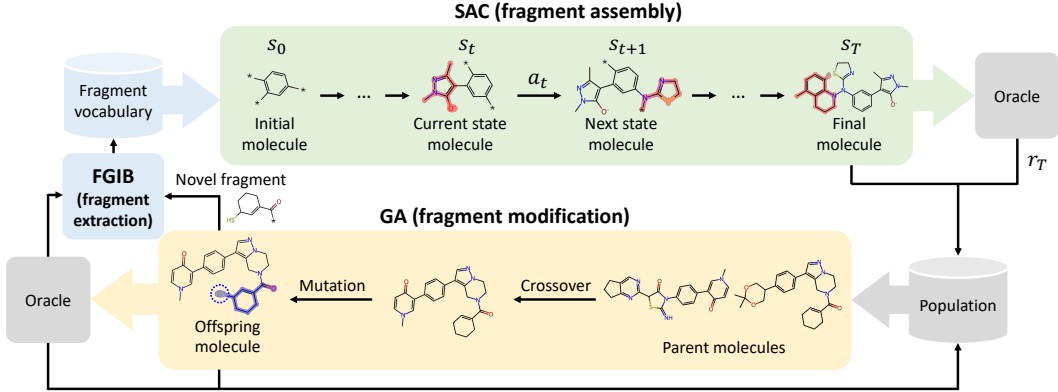

Figure 2: **The overall framework of GEAM.** GEAM consists of three modules, FGIB, SAC, and GA for fragment extraction, fragment assembly, and fragment modification, respectively.

and Maziarz et al. (2022) proposed to obtain fragments by breaking some of the bonds with a predefined rule (e.g., acyclic sing bonds), then select the most frequent fragments. Kong et al. (2022) and Geng et al. (2023) utilized merge-and-update rules to find the frequent fragments in the given molecules. All of these methods do not consider the target properties. On the other hand, Jin et al. (2020b) proposed to find molecular substructures that satisfy the given properties, but the approach requires an expensive oracle call to examine each building block candidate in a brute-force manner, and the substructures are not actually fragments in that they are already full molecules that have chemical properties and are not assembled together. Consequently, the found substructures are large in size and often few in number, resulting in low novelty and diversity of the generated molecules.

**Fragment-based molecule generation** Fragment-based molecular generative models denote the models that use the extracted fragments as building blocks and learn to assemble the blocks into molecules. Xie et al. (2020) proposed to use MCMC sampling when assemble or delete the fragments. Yang et al. (2021) proposed to use a reinforcement learning (RL) model and view fragment addition as actions. Maziarz et al. (2022), Kong et al. (2022) and Geng et al. (2023) proposed to use a VAE to assemble the fragments. The model of Jin et al. (2020b) learns to complete the obtained molecular substructures into final molecules by adding molecular branches.

**Subgraph recognition** Given a graph, subgraph recognition aims to find a compressed subgraph that contains salient information to predict the property of the graph. Graph information bottleneck (GIB) (Wu et al., 2020) approached this problem by considering the subgraph as a bottleneck random variable and applying the information bottleneck theory. Yu et al. (2022) proposed to utilize Gaussian noise injection into node representations to confine the information and recognize important subgraphs, while Miao et al. (2022) proposed to consider the subgraph attention process as the information bottleneck. Lee et al. (2023a) applied the GIB principle to molecular relational learning tasks. In practice, it is common for these methods to recognize disconnected substructures rather than connected fragments. Subgraph recognition by GIB has been only employed in classification and regression tasks, and this is the first work that applies GIB to fragment extraction.

## 3 METHOD

We now introduce our Goal-aware fragment Extraction, Assembly, and Modification (GEAM) framework which aims to generate molecules that satisfy the target properties with goal-aware fragments. We first describe the goal-aware fragment extraction method in Section 3.1. Then we describe the fragment assembly method in Section 3.2. Finally, we describe the fragment modification method, the dynamic vocabulary update, and the resulting GEAM in Section 3.3.

### 3.1 GOAL-AWARE FRAGMENT EXTRACTION

Assume that we are given a set of $N$ molecular graphs $G_i$ with its corresponding properties $Y_i \in [0, 1]$, denoted as $\mathcal{D} = \{(G_i, Y_i)\}_{i=1}^{N}$. Each graph $G_i = (\boldsymbol{X}_i, \boldsymbol{A}_i)$ consists of $n$ nodes with a node feature matrix $\boldsymbol{X}_i \in \mathbb{R}^{n \times d}$ and an adjacency matrix $\boldsymbol{A}_i \in \mathbb{R}^{n \times n}$. Let $\mathcal{V}$ be a set of all nodes from

the graphs $\mathcal{G} = \{G_i\}_{i=1}^N$ and let $\mathcal{E}$ be a set of all edges from $\mathcal{G}$. Our goal is to extract goal-aware fragments from $\mathcal{G}$ such that we can assemble these fragments to synthesize graphs with desired properties. In order to achieve this goal, we propose Fragment-wise Graph Information Bottleneck (FGIB), a model that learns to identify salient fragments of $G_i$ for predicting the target property $Y_i$.

Concretely, we first decompose a set of the graphs $\mathcal{G}$ into $M$ candidate fragments, denoted as $\mathcal{F}$ with BRICS (Degen et al., 2008), a popular method that fragmentizes molecules into retrosynthetically interesting substructures. Each fragment $F = (V, E) \in \mathcal{F}$ is comprised of vertices $V \subset \mathcal{V}$ and edges $E \subset \mathcal{E}$. Then each graph $G$ can be represented as $m$ fragments, $\{F_j = (V_j, E_j)\}_{j=1}^m$, with $F_j \in \mathcal{F}$. Inspired by graph information bottleneck (Wu et al., 2020), FGIB identifies a subgraph $G^{\text{sub}}$ that is maximally informative for predicting the target property $Y$ while maximally compressing the original graph $G$:

$$\min_{G^{\text{sub}}} -I(G^{\text{sub}}, Y) + \beta I(G^{\text{sub}}, G), \tag{1}$$

where $\beta > 0$ and $I(X, Y)$ denotes the mutual information between the random variables $X$ and $Y$.

FGIB first calculates the node embeddings $\{\mathbf{h}\}_{i=1}^n$ from the graph $G$ with an MPNN (Gilmer et al., 2017) and use average pooling to obtain the fragment embedding $\mathbf{e}_j$ of the fragment $F_j$ as follows:

$$[\mathbf{h}_1 \cdots \mathbf{h}_n]^\top = \text{MPNN}(\boldsymbol{X}, \boldsymbol{A}), \quad \mathbf{e}_j = \text{AvgPool}(\{\mathbf{h}_l : v_l \in V_j\}) \in \mathbb{R}^d, \tag{2}$$

where $v_l$ denotes the node whose corresponding node embedding is $\mathbf{h}_l$. Using an MLP with a sigmoid activation function, we obtain $w_j \in [0, 1]$, the importance of the fragment $F_j$ for predicting the target property $Y$, as $w_j = \text{MLP}(\mathbf{e}_j)$. We denote $\theta$ as the parameters of the MPNN and the MLP. Following Yu et al. (2022), we inject a noise to the fragment embedding $\mathbf{e}_j$ according to $w_j$ to control the information flow from $G$ as follows:

$$\tilde{\mathbf{e}}_j = w_j \mathbf{e}_j + (1 - w_j)\hat{\boldsymbol{\mu}}_j + \boldsymbol{\epsilon}, \quad w_j = \text{MLP}(\mathbf{e}_j), \quad \boldsymbol{\epsilon} \sim \mathcal{N}(\mathbf{0}, (1 - w_j)\hat{\boldsymbol{\Sigma}}), \tag{3}$$

where $\hat{\boldsymbol{\mu}}_j \in \mathbb{R}^d$ and $\hat{\boldsymbol{\Sigma}} \in \mathbb{R}^{d \times d}$ denote an empirical mean vector and a diagonal covariance matrix estimated from $\{\mathbf{e}_j\}_{j=1}^m$, respectively. Intuitively, the more a fragment is considered to be irrelevant for predicting the target property (i.e., small weight $w$), the more the transmission of the fragment information is blocked. Let $Z = \text{vec}([\tilde{\mathbf{e}}_1 \cdots \tilde{\mathbf{e}}_m])$ be the embedding of the perturbed fragments, which is a Gaussian-distributed random variable, i.e., $p_\theta(Z|G) = \mathcal{N}(\boldsymbol{\mu}_\theta(G), \boldsymbol{\Sigma}_\theta(G))$. Here vec denotes a vectorization of a matrix, and $\boldsymbol{\mu}_\theta(G)$ and $\boldsymbol{\Sigma}_\theta(G)$ denote the mean and the covariance induced by the MPNN and the MLP with the noise $\boldsymbol{\epsilon}$, respectively. Assuming that there is no information loss in the fragments after encoding them, our objective function in Eq. (1) becomes optimization the parameters $\theta$ such that we can still predict the property $Y$ from the perturbed fragment embedding $Z$ while minimizing the mutual information between $G$ and $Z$ as follows:

$$\min_\theta \underbrace{-I(Z, Y; \theta) + \beta I(Z, G; \theta)}_{\mathcal{L}_{\text{IB}}(\theta)} \tag{4}$$

Following Alemi et al. (2017), we can derive the upper bound of $\mathcal{L}_{\text{IB}}(\theta)$ with variational inference:

$$\mathcal{L}(\theta, \phi) \coloneqq \frac{1}{N} \sum_{i=1}^N \left( -\log q_\phi(Y_i|Z_i) + \beta D_{\text{KL}}(p_\theta(Z|G_i) \,\|\, u(Z)) \right), \tag{5}$$

where $q_\phi$ is a property predictor that takes the perturbed fragment embedding $Z$ as an input, $u(Z)$ is a variational distribution that approximates the marginal $p_\theta(Z)$, and $Z_i$ is drawn from $p_\theta(Z|G_i) = \mathcal{N}(\boldsymbol{\mu}_\theta(G_i), \boldsymbol{\Sigma}_\theta(G_i))$ for $i \in \{1, \ldots, N\}$. We optimize $\theta$ and $\phi$ to minimize the objective function $\mathcal{L}(\theta, \phi)$. Note that the variational distribution $u(\cdot)$ is chosen to be Gaussian with respect to $Z$, enabling analytic computation of the KL divergence. A detail proof is included in Appendix B.

After training FGIB, we score each fragment $F_j = (V_j, E_j) \in \mathcal{F}$ with FGIB as follows:

$$\text{score}(F_j) = \frac{1}{|S(F_j)|} \sum_{(G,Y) \in S(F_j)} \frac{w_j(G, F_j)}{\sqrt{|V_j|}} \cdot Y \in [0, 1], \tag{6}$$

where $S(F_j) = \{(G, Y) \in \mathcal{D} : F_j \text{ is a subgraph of } G\}$ and $w_j(G, F_j)$ is an importance of the fragment $F_j$ in the graph $G$, computed as Eq. (3). Intuitively, the score quantifies the extent to which a fragment contributes to achieving a high target property. Specifically, the term $w_j(G, F_j)/\sqrt{|V_j|}$

measures how much a fragment contributes to its whole molecule in terms of the target property, while the term $Y$ measures the property of the molecule. As the number of nodes of the fragment becomes larger, FGIB is more likely to consider it important when predicting the property. In order to normalize the effect of the fragment size, we include $\sqrt{|V_j|}$ in the first term. Based on the scores of all fragments, we choose the top-$K$ fragments as the goal-aware vocabulary $\mathcal{S} \subset \mathcal{F}$ for the subsequent generation of molecular graphs with desired properties.

## 3.2 FRAGMENT ASSEMBLY

The next step is to generate molecules with the extracted goal-aware fragment vocabulary. For generation, we introduce the fragment assembly module, which is a soft-actor critic (SAC) model that learns to assemble the fragments to generate molecules with desired properties.

We formulate fragment assembly as an RL problem, following Yang et al. (2021). Given a partially generated molecule $g_t$ which becomes a state $\mathbf{s}_t$ at time step $t$, a policy network adds a fragment to $g_t$ by sequentially selecting three actions: 1) the attachment site of $g_t$ to use in forming a new bond, 2) the fragment $F \in \mathcal{S}$ to be attached to $g_t$, and 3) the attachment site of $F$ to use in forming a new bond. Following Yang et al. (2021), we encode the nodes of the graph $g_t$ with a GCN (Kipf & Welling, 2017) as $\boldsymbol{H} = \mathrm{GCN}(g_t)$ and obtain the graph embedding with sum pooling as $\mathbf{h}_{g_t} = \mathrm{SumPool}(\boldsymbol{H})$. Given $\boldsymbol{H}$ and $\mathbf{h}_{g_t}$, we parameterize the policy network $\pi$ with three sub-policy networks to sequentially choose actions conditioned on previous ones:

$$p_{\pi_1}(\cdot|\mathbf{s}_t) = \pi_1(\boldsymbol{Z}_1), \ \boldsymbol{Z}_1 = [\mathbf{z}_{1,1} \cdots \mathbf{z}_{1,n_1}]^\top = f_1(\mathbf{h}_{g_t}, \boldsymbol{H}_{\mathrm{att}}) \tag{7}$$

$$p_{\pi_2}(\cdot|a_1, \mathbf{s}_t) = \pi_2(\boldsymbol{Z}_2), \ \boldsymbol{Z}_2 = [\mathbf{z}_{2,1} \cdots \mathbf{z}_{2,n_2}]^\top = f_2(\mathbf{z}_{1,a_1}, \mathrm{ECFP}(\mathcal{S})) \tag{8}$$

$$p_{\pi_3}(\cdot|a_1, a_2, \mathbf{s}_t) = \pi_3(\boldsymbol{Z}_3), \ \boldsymbol{Z}_3 = [\mathbf{z}_{3,1} \cdots \mathbf{z}_{3,n_3}]^\top = f_3(\mathrm{SumPool}(\mathrm{GCN}(F_{a_2})), \boldsymbol{H}_{\mathrm{att}, F_{a_2}}), \tag{9}$$

where $\boldsymbol{H}_{\mathrm{att}}$ denotes the node embeddings of the attachment sites. We employ multiplicative interactions (Jayakumar et al., 2020) for $f_1$, $f_2$ and $f_3$ to fuse two inputs from heterogeneous spaces. The first policy network $\pi_1$ outputs categorical distribution over attachment sites of the current graph $g_t$ conditioned on $\mathbf{h}_{g_t}$ and $\boldsymbol{H}_{\mathrm{att}}$, and chooses the attachment site with $a_1 \sim p_{\pi_1}(\cdot|\mathbf{s}_t)$. The second policy network $\pi_2$ selects the fragment $F_{a_2} \in \mathcal{S}$ with $a_2 \sim p_{\pi_2}(\cdot|a_1, \mathbf{s}_t)$, conditioned on the embedding of the previously chosen attachment site $\mathbf{z}_{1,a_1}$ and the ECFPs of all the fragments $\mathrm{ECFP}(\mathcal{S})$. Then we encode the node embeddings of the fragment $F_{a_2}$ with the same GCN as $\boldsymbol{H}_{F_{a_2}} = \mathrm{GCN}(F_{a_2})$, and get the fragment embedding $\mathbf{h}_{F_{a_2}} = \mathrm{SumPool}(\boldsymbol{H}_{F_{a_2}})$. The policy network $\pi_3$ chooses the attachment site of the fragment $F_{a_2}$ with $a_3 \sim p_{\pi_3}(\cdot|a_1, a_2, \mathbf{s}_t)$, conditioned on the fragment embedding $\mathbf{h}_{F_{a_2}}$ and the attachment site embeddings of the fragment $\boldsymbol{H}_{\mathrm{att}, F_{a_2}}$. Finally, we attach the fragment $F_{a_2}$ to the current graph $g_t$ with the chosen attachment sites $a_1$ and $a_3$, resulting in a new graph $g_{t+1}$. With $T$ steps of sampling actions $(a_1, a_2, a_3)$ using the policy network, we generate a new molecule $g_T = G$, call the oracle to evaluate the molecule $G$ and calculate the reward $r_T$.

With the SAC objective (Haarnoja et al., 2018), we train the policy network $\pi$ as follows:

$$\pi^* = \arg\max_\pi \sum_t \mathbb{E}_{(\mathbf{s}_t, \mathbf{a}_t) \sim \rho_\pi}[r(\mathbf{s}_t, \mathbf{a}_t) + \alpha \mathcal{H}(\pi(\cdot|\mathbf{s}_t))], \tag{10}$$

where $r(\mathbf{s}_t, \mathbf{a}_t)$ is a reward function[1], $\mathcal{H}(\pi(\cdot|\mathbf{s}_t))$ is entropy of action probabilities given $\mathbf{s}_t$ with a temperature parameter $\alpha > 0$, and $\rho_\pi(\mathbf{s}_t, \mathbf{a}_t)$ is a state-action marginal of the trajectory distribution induced by the policy $\pi(\mathbf{a}_t|\mathbf{s}_t) = p_{\pi_3}(a_{3,t}|a_{2,t}, a_{1,t}, \mathbf{s}_t) \cdot p_{\pi_2}(a_{2,t}|a_{t,1}, \mathbf{s}_t) \cdot p_{\pi_1}(a_{t,1}|\mathbf{s}_t)$ with $\mathbf{a}_t = (a_{1,t}, a_{2,t}, a_{3,t})$. In order to sample discrete actions differentiable for backpropagation, we use Gumbel-Softmax (Jang et al., 2017; Maddison et al., 2017) to optimize Eq. (10).

## 3.3 FRAGMENT MODIFICATION AND DYNAMIC VOCABULARY UPDATE

With the fragment assembly module only, we are unable to generate molecules consisting of fragments not included in the predefined vocabulary, which hinders generation of diverse molecules and precludes exploration beyond the vocabulary. In order to overcome this problem, we introduce the fragment modification module, which utilizes a genetic algorithm (GA) to generate molecules that contain novel fragments.

---

[1]We set the intermediate rewards to 0.05, so that only final molecules are evaluated by the oracle.

Specifically, we employ a graph-based genetic algorithm (GA) (Jensen, 2019). At the first round of the GA, we initialize the population with the top-$P$ molecules generated by the fragment assembly module. The GA then selects parent molecules from the population and generates offspring molecules by performing crossover and mutation. As a consequence of the crossover and mutation operations, the generated offspring molecules contain novel fragments not in the initial vocabulary. In the subsequent rounds, we choose the top-$P$ molecules generated so far by both SAC and GA to construct the GA population of the next round.

We iteratively run the fragment assembly module described in Section 3.2 and the fragment modification in turn, and this generative scheme is referred to as GEAM-static. To further enhance molecular diversity and novelty, we propose incorporating the fragment extraction module into this generative cycle. Concretely, in each cycle after the fragment assembly and the fragment modification modules generate molecules, FGIB extracts novel goal-aware fragments $\mathcal{S}'$ from the offspring molecules as described in Section 3.1. Then the vocabulary is dynamically updated as $\mathcal{S} \cup \mathcal{S}'$. When the size of the vocabulary becomes larger than the maximum size $L$, we choose the top-$L$ fragments as the vocabulary based on the scores in Eq. (6). The fragment assembly module assembles fragments of the updated vocabulary in the next iteration, and we refer to this generative framework as GEAM. The single generation cycle of GEAM is described in Algorithm 1 in Section A.

## 4 EXPERIMENTS

We demonstrate the efficacy of our proposed GEAM in two sets of multi-objective molecular optimization tasks that simulate real-world drug discovery problems. We first conduct the experiment to generate novel molecules that have high binding affinity, drug-likeness, and synthesizability in Section 4.1. We then experiment on the practical molecular optimization (PMO) benchmark in Section 4.2. We further conduct extensive ablation studies and qualitative analysis in Section 4.3.

### 4.1 OPTIMIZATION OF BINDING AFFINITY UNDER QED, SA AND NOVELTY CONSTRAINTS

**Experimental setup**   Following Lee et al. (2023b), we validate GEAM in the five docking score (DS) optimization tasks under the quantitative estimate of drug-likeness (QED) (Bickerton et al., 2012), synthetic accessibility (SA) (Ertl & Schuffenhauer, 2009), and novelty constraints. In these tasks, the goal is to generate novel, drug-like, and synthesizable molecules that have a high absolute value of the docking score. Following Lee et al. (2023b), we set the property $Y$ as follows:

$$Y(G) = \widehat{\mathrm{DS}}(G) \times \mathrm{QED}(G) \times \widehat{\mathrm{SA}}(G) \in [0, 1], \qquad (11)$$

where $\widehat{\mathrm{DS}}$ and $\widehat{\mathrm{SA}}$ are the normalized DS and the normalized SA, respectively (Eq. (16)). We use ZINC250k (Irwin et al., 2012) to train FGIB to predict $Y$ and extract initial fragments. Optimization performance is evaluated with 3,000 generated molecules using the following metrics. **Novel hit ratio (%)** measures the fraction of unique and novel hits among the generated molecules. Here, *novel* molecules is defined as the molecules that have the maximum Tanimoto similarity less than $0.4$ with the molecules in the training set, and *hit* is the molecules that satisfy the following criteria: DS $<$ (the median DS of known active molecules), QED $> 0.5$ and SA $< 5$. **Novel top 5% DS (kcal/mol)** measures the average DS of the top 5% unique, novel hits. parp1, fa7, 5ht1b, braf and jak2 are used as the protein targets the docking scores are calculated for. In addition, we evaluate the fraction of novel molecules, **novelty (%)**, and the extent of chemical space covered, **#Circles** (Xie et al., 2023) of the generated hits. The details are provided in Section C.1 and Section C.2.

**Baselines**   **REINVENT** (Olivecrona et al., 2017) is a SMILES-based RL model with a pretrained prior. **Graph GA** (Jensen, 2019) is a GA-based model that utilizes predefined crossover and mutation rules. **MORLD** (Jeon & Kim, 2020) is an RL model that uses the MolDQN algorithm (Zhou et al., 2019). **HierVAE** (Jin et al., 2020a) is a VAE-based model that uses the hierarchical motif representation of molecules. **RationaleRL** (Jin et al., 2020b) is an RL model that first identifies subgraphs that are likely responsible for the target properties (i.e., rationale) and then extends those to complete molecules. **FREED** (Yang et al., 2021) is an RL model that assembles the fragments obtained using CReM (Polishchuk, 2020). **PS-VAE** (Kong et al., 2022) is a VAE-based model that uses the mined principal subgraphs as the building blocks. **MOOD** (Lee et al., 2023b) is a diffusion model that incorporates an out-of-distribution (OOD) control to enhance novelty. The details are provided in Section C.2, and the results of additional baselines are included in Table 7 and Table 8.

Table 1: **Novel hit ratio (%) results.** The results are the means and the standard deviations of 3 runs. The results for the baselines except for RationaleRL and PS-VAE are taken from Lee et al. (2023b). The best results are highlighted in bold.

| Method | Target protein | | | | |
|---|---|---|---|---|---|
| | parp1 | fa7 | 5ht1b | braf | jak2 |
| REINVENT (Olivecrona et al., 2017) | 0.480 (± 0.344) | 0.213 (± 0.081) | 2.453 (± 0.561) | 0.127 (± 0.088) | 0.613 (± 0.167) |
| Graph GA (Jensen, 2019) | 4.811 (± 1.661) | 0.422 (± 0.193) | 7.011 (± 2.732) | 3.767 (± 1.498) | 5.311 (± 1.667) |
| MORLD (Jeon & Kim, 2020) | 0.047 (± 0.050) | 0.007 (± 0.013) | 0.880 (± 0.735) | 0.047 (± 0.040) | 0.227 (± 0.118) |
| HierVAE (Jin et al., 2020a) | 0.553 (± 0.214) | 0.007 (± 0.013) | 0.507 (± 0.278) | 0.207 (± 0.220) | 0.227 (± 0.127) |
| RationaleRL (Jin et al., 2020b) | 4.267 (± 0.450) | 0.900 (± 0.098) | 2.967 (± 0.307) | 0.000 (± 0.000) | 2.967 (± 0.196) |
| FREED (Yang et al., 2021) | 4.627 (± 0.727) | 1.332 (± 0.113) | 16.767 (± 0.897) | 2.940 (± 0.359) | 5.800 (± 0.295) |
| PS-VAE (Kong et al., 2022) | 1.644 (± 0.389) | 0.478 (± 0.140) | 12.622 (± 1.437) | 0.367 (± 0.047) | 4.178 (± 0.933) |
| MOOD (Lee et al., 2023b) | 7.017 (± 0.428) | 0.733 (± 0.141) | 18.673 (± 0.423) | 5.240 (± 0.285) | 9.200 (± 0.524) |
| GEAM-static (ours) | 39.667 (± 4.493) | 16.989 (± 1.959) | 38.433 (± 2.103) | 27.422 (± 0.494) | 42.056 (± 1.855) |
| GEAM (ours) | **40.567** (± 0.825) | **20.711** (± 1.873) | **38.489** (± 0.350) | **27.900** (± 1.822) | **42.950** (± 1.117) |

Table 2: **Novel top 5% docking score (kcal/mol) results.** The results are the means and the standard deviations of 3 runs. The results for the baselines except for RationaleRL and PS-VAE are taken from Lee et al. (2023b). The best results are highlighted in bold.

| Method | Target protein | | | | |
|---|---|---|---|---|---|
| | parp1 | fa7 | 5ht1b | braf | jak2 |
| REINVENT (Olivecrona et al., 2017) | -8.702 (± 0.523) | -7.205 (± 0.264) | -8.770 (± 0.316) | -8.392 (± 0.400) | -8.165 (± 0.277) |
| Graph GA (Jensen, 2019) | -10.949 (± 0.532) | -7.365 (± 0.326) | -10.422 (± 0.670) | -10.789 (± 0.341) | -10.167 (± 0.576) |
| MORLD (Jeon & Kim, 2020) | -7.532 (± 0.260) | -6.263 (± 0.165) | -7.869 (± 0.650) | -8.040 (± 0.337) | -7.816 (± 0.133) |
| HierVAE (Jin et al., 2020a) | -9.487 (± 0.278) | -6.812 (± 0.274) | -8.081 (± 0.252) | -8.978 (± 0.525) | -8.285 (± 0.370) |
| RationaleRL (Jin et al., 2020b) | -10.663 (± 0.086) | -8.129 (± 0.048) | -9.005 (± 0.155) | *No hit found* | -9.398 (± 0.076) |
| FREED (Yang et al., 2021) | -10.579 (± 0.104) | -8.378 (± 0.044) | -10.714 (± 0.183) | -10.561 (± 0.080) | -9.735 (± 0.022) |
| PS-VAE (Kong et al., 2022) | -9.978 (± 0.091) | -8.028 (± 0.050) | -9.887 (± 0.115) | -9.637 (± 0.049) | -9.464 (± 0.129) |
| MOOD (Lee et al., 2023b) | -10.865 (± 0.113) | -8.160 (± 0.071) | -11.145 (± 0.042) | -11.063 (± 0.034) | -10.147 (± 0.060) |
| GEAM-static (ours) | -12.810 (± 0.124) | -9.682 (± 0.026) | -12.369 (± 0.084) | -12.336 (± 0.157) | -11.812 (± 0.055) |
| GEAM (ours) | **-12.891** (± 0.158) | **-9.890** (± 0.116) | **-12.374** (± 0.036) | **-12.342** (± 0.095) | **-11.816** (± 0.067) |

Table 3: **Novelty (%) results.** The results are the means and the standard deviations of 3 runs. The results for the baselines except for RationaleRL and PS-VAE are taken from Lee et al. (2023b). The best results are highlighted in bold.

| Method | Target protein | | | | |
|---|---|---|---|---|---|
| | parp1 | fa7 | 5ht1b | braf | jak2 |
| REINVENT (Olivecrona et al., 2017) | 9.894 (± 2.178) | 10.731 (± 1.516) | 11.605 (± 3.688) | 8.715 (± 2.712) | 11.456 (± 1.793) |
| MORLD (Jeon & Kim, 2020) | **98.433** (± 1.189) | **97.967** (± 1.764) | **98.787** (± 0.743) | **96.993** (± 2.787) | **97.720** (± 0.995) |
| HierVAE (Jin et al., 2020a) | 60.453 (± 17.165) | 24.853 (± 15.416) | 48.107 (± 1.988) | 59.747 (± 16.403) | 85.200 (± 14.262) |
| RationaleRL (Jin et al., 2020b) | 9.300 (± 0.354) | 9.802 (± 0.166) | 7.133 (± 0.141) | 0.000 (± 0.000) | 7.389 (± 0.220) |
| FREED (Yang et al., 2021) | 74.640 (± 2.953) | 78.787 (± 2.132) | 75.027 (± 5.194) | 73.653 (± 4.312) | 75.907 (± 5.916) |
| PS-VAE (Kong et al., 2022) | 60.822 (± 2.251) | 56.611 (± 1.892) | 57.956 (± 2.181) | 57.744 (± 2.710) | 58.689 (± 2.307) |
| MOOD (Lee et al., 2023b) | 84.180 (± 2.123) | 83.180 (± 1.519) | 84.613 (± 0.822) | 87.413 (± 0.830) | 83.273 (± 1.455) |
| GEAM-static (ours) | 84.344 (± 5.290) | 86.144 (± 6.807) | 79.389 (± 3.903) | 87.122 (± 2.163) | 86.633 (± 1.817) |
| GEAM (ours) | 88.611 (± 3.107) | 89.378 (± 2.619) | 84.222 (± 2.968) | 90.322 (± 3.467) | 89.222 (± 1.824) |

Table 4: **#Circles of generated hit molecules.** The #Circles threshold is set to 0.75. The results are the means and the standard deviations of 3 runs. The results for the baselines except for RationaleRL and PS-VAE are taken from Lee et al. (2023b). The best results are highlighted in bold.

| Method | Target protein | | | | |
|---|---|---|---|---|---|
| | parp1 | fa7 | 5ht1b | braf | jak2 |
| REINVENT (Olivecrona et al., 2017) | 44.2 (± 15.5) | 23.2 (± 6.6) | 138.8 (± 19.4) | 18.0 (± 2.1) | 59.6 (± 8.1) |
| MORLD (Jeon & Kim, 2020) | 1.4 (± 1.5) | 0.2 (± 0.4) | 22.2 (± 16.1) | 1.4 (± 1.2) | 6.6 (± 3.7) |
| HierVAE (Jin et al., 2020a) | 4.8 (± 1.6) | 0.8 (± 0.7) | 5.8 (± 1.0) | 3.6 (± 1.4) | 4.8 (± 0.7) |
| RationaleRL (Jin et al., 2020b) | 61.3 (± 1.2) | 2.0 (± 0.0) | **312.7** (± 6.3) | 1.0 (± 0.0) | **199.3** (± 7.1) |
| FREED (Yang et al., 2021) | 34.8 (± 4.9) | 21.2 (± 4.0) | 88.2 (± 13.4) | 34.4 (± 8.2) | 59.6 (± 8.2) |
| PS-VAE (Kong et al., 2022) | 38.0 (± 6.4) | 18.0 (± 5.9) | 180.7 (± 11.6) | 16.0 (± 0.8) | 83.7 (± 11.9) |
| MOOD (Lee et al., 2023b) | 86.4 (± 11.2) | 19.2 (± 4.0) | 144.4 (± 15.1) | 50.8 (± 3.8) | 81.8 (± 5.7) |
| GEAM-static (ours) | 114.0 (± 2.9) | 60.7 (± 4.0) | 134.7 (± 8.5) | 70.0 (± 2.2) | 99.3 (± 1.7) |
| GEAM (ours) | **123.0** (± 7.8) | **79.0** (± 9.2) | 144.3 (± 8.6) | **84.7** (± 8.6) | 118.3 (± 0.9) |

**Results** The results are shown in Table 1 and Table 2. GEAM and GEAM-static significantly outperform all the baselines in all the tasks, demonstrating that the proposed goal-aware extraction method and the proposed combination of SAC and GA are highly effective in discovering novel, drug-like, and synthesizable drug candidates that have high binding affinity. GEAM shows comparable or better performance than GEAM-static, and as shown in Table 3 and Table 4, the usage of the dynamic vocabulary update enhances novelty and diversity without degrading optimization

Table 5: **PMO MPO AUC top-100 results.** The results are the means of 3 runs. The results for the baselines are taken from Gao et al. (2022). The best results are highlighted in bold.

| Method | Benchmark | | | | | | | Average |
|---|---|---|---|---|---|---|---|---|
| | Amlodipine | Fexofenadine | Osimertinib | Perindopril | Ranolazine | Sitagliptin | Zaleplon | |
| REINVENT (Olivecrona et al., 2017) | 0.608 | 0.752 | 0.806 | 0.511 | 0.719 | 0.006 | 0.325 | 0.532 |
| Graph GA (Jensen, 2019) | 0.622 | 0.731 | 0.799 | 0.503 | 0.670 | 0.330 | 0.305 | 0.566 |
| STONED (Nigam et al., 2021) | 0.593 | 0.777 | 0.799 | 0.472 | **0.738** | 0.351 | 0.307 | 0.577 |
| GEAM-static (ours) | 0.602 | 0.796 | 0.828 | 0.501 | 0.703 | 0.346 | 0.397 | 0.596 |
| GEAM (ours) | **0.626** | **0.799** | **0.831** | **0.514** | 0.714 | **0.417** | **0.402** | **0.615** |

Table 6: **PMO MPO novelty (%) / #Circles results.** The #Circles threshold is set to 0.75. The results are the means of 3 runs. The best results are highlighted in bold.

| Method | Benchmark | | | | | | |
|---|---|---|---|---|---|---|---|
| | Amlodipine | Fexofenadine | Osimertinib | Perindopril | Ranolazine | Sitagliptin | Zaleplon |
| REINVENT (Olivecrona et al., 2017) | 17.0 / 303.7 | 13.4 / 343.3 | 25.0 / **452.3** | 33.1 / 318.3 | 15.6 / 253.3 | 15.7 / **398.3** | 7.6 / 275.3 |
| Graph GA (Jensen, 2019) | 61.1 / 258.7 | 76.2 / 333.3 | 64.1 / 270.3 | 44.4 / 278.7 | 78.2 / **364.7** | 88.0 / 306.3 | 41.3 / 272.7 |
| STONED (Nigam et al., 2021) | 82.7 / 303.7 | 91.6 / 330.3 | 88.1 / 301.3 | 65.8 / 301.0 | **92.4** / 316.7 | **89.5** / 326.3 | 63.1 / 280.3 |
| GEAM-static (ours) | 83.1 / 412.0 | 97.6 / 397.7 | 94.5 / 315.3 | 93.2 / 318.0 | 68.9 / 256.7 | 73.7 / 233.0 | 76.2 / 267.0 |
| GEAM (ours) | **84.2 / 424.0** | **98.0 / 502.0** | **97.0** / 435.0 | **95.3 / 377.3** | 82.7 / 295.3 | 86.9 / 257.0 | **81.7 / 336.0** |

performance. There is a general trend that the more powerful the molecular optimization model, the less likely it is to generate diverse molecules (Gao et al., 2022), but GEAM effectively overcomes this trade-off by discovering novel and high-quality goal-aware fragments on-the-fly. Note that the high novelty values of MORLD are trivial due to its poor optimization performance and very low diversity. In the same vein, the high diversity values of RationaleRL on the target proteins 5ht1b and jak2 are not meaningful due to its poor optimization performance and novelty.

## 4.2 OPTIMIZATION OF MULTI-PROPERTY OBJECTIVES IN PMO BENCHMARK

**Experimental setup** We validate GEAM in the seven multi-property objective (MPO) optimization tasks in the practical molecular optimization (PMO) benchmark (Gao et al., 2022), which are the tasks in the Guacamol benchmark (Brown et al., 2019) that additionally take the number of oracle calls into account for realistic drug discovery. The details are provided in Section C.1 and C.3.

**Baselines** We use the top three models reported by Gao et al. (2022) as our baselines. In addition to **REINVENT** (Olivecrona et al., 2017) and **Graph GA** (Jensen, 2019), **STONED** (Nigam et al., 2021) is a GA-based model that manipulates SELFIES strings.

**Results** As shown in Table 5, GEAM outperform the baselines in most of the tasks, demonstrating its applicability to various drug discovery problems. Note that GEAM distinctly improves the performance of GEAM-static in some tasks. Furthermore, as shown in Table 6, GEAM shows higher novelty and diversity than other methods. Especially, GEAM generates more novel and diverse molecules than GEAM-static, again verifying the dynamic vocabulary update of GEAM effectively improves novelty and diversity without degrading optimization performance.

## 4.3 ABLATION STUDIES AND QUALITATIVE ANALYSIS

**Effect of the goal-aware fragment extraction** To examine the effect of the proposed goal-aware fragment extraction method with FGIB, in Figure 3(a), we compare FREED with FREED (FGIB), which is a variant of FREED that uses the fragment vocabulary extracted by FGIB as described in Section 3.1. FREED (FGIB) outperforms FREED by a large margin, indicating the proposed goal-aware fragment extraction method with FGIB largely boosts the optimization performance. We also compare GEAM against GEAM with different fragment vocabularies in Figure 3(b). GEAM (FREED), GEAM (MiCaM), GEAM (BRICS) are the GEAM variants that use the FREED vocabulary, the MiCaM (Geng et al., 2023) vocabulary, the random BRICS (Degen et al., 2008) vocabulary, respectively. GEAM (property) is GEAM which only uses the property instead of Eq. (6) when scoring fragments, i.e., $\text{score}(F_j) = \frac{1}{|S(F_j)|} \sum_{(G,Y) \in S(F_j)} Y$. GEAM significantly outperforms all the variants, verifying the importance of our goal-aware fragment vocabulary. Notably, GEAM (property) uses the topmost fragments in terms of the target property, but performs worse than GEAM because it does not use FGIB to find important subgraphs that contribute to the property.

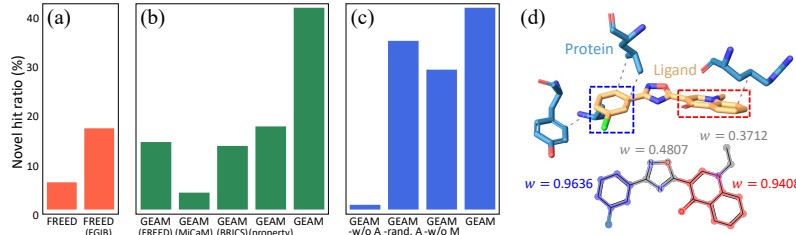

Figure 3: (a-c) **Ablation studies on FGIB, SAC and GA** on the ligand generation task with the target protein jak2 and (d) **the PLIP image** showing hydrophobic interactions between an example molecule and jak2.

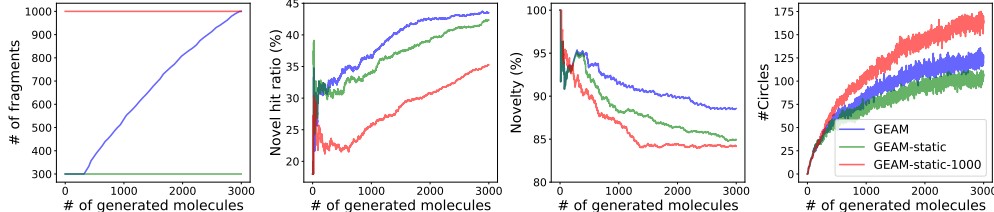

Figure 4: **The generation progress of GEAM and GEAM-static** on the ligand generation task against jak2.

**Effect of the fragment assembly and modification**    To examine the effect of the proposed combinatorial use of the assembly and the modification modules, we compare GEAM with GEAM-w/o A and GEAM-w/o M in Figure 3(c). GEAM-w/o A does not use the assembly module and constructs its population as the top-$P$ molecules from ZINC250k, while GEAM-w/o M does not use the modification module. GEAM-random A uses random fragment assembly instead of SAC. We can observe GEAM-w/o A significantly underperforms as the fragment modification module alone cannot take the advantage of the goal-aware fragments, and GEAM-random A largely improves over GEAM-w/o A. GEAM outperforms all the ablated variants, demonstrating that jointly leveraging the fragment assembly module and the fragment modification module is crucial to the performance.

**Effect of the dynamic vocabulary update**    To thoroughly examine the effect of the proposed dynamic update of the fragment vocabulary, we compare the generation progress of GEAM with that of GEAM-static in Figure 4. GEAM-static-1000 is GEAM-static with the vocabulary size $K = 1,000$. When the initial vocabulary size $K = 300$ and the maximum vocabulary size $L = 1,000$, the vocabulary size of GEAM increases during generation from 300 to 1,000 as GEAM dynamically collects fragments on-the-fly while the vocabulary sizes of GEAM-static and GEAM-static-1000 are fixed to 300 and 1,000, respectively. As expected, GEAM-static-1000 shows the worst optimization performance since its vocabulary consists of top-1,000 fragments instead of top-300 from the same training molecules, and shows the highest diversity as it utilizes more fragments than GEAM and GEAM-static throughout the generation process. GEAM shows the best optimization performance and novelty thanks to its vocabulary update strategy that constantly incorporates novel fragments outside the training molecules, as well as improved diversity compared to GEAM-static.

**Qualitative analysis**    We qualitatively analyze the extracted goal-aware fragments. In Figure 3(d), we present an example of the binding interactions of a molecule and the target protein jak2 using the protein-ligand interaction profiler (PLIP) (Adasme et al., 2021). Additionally, we show the fragments of the molecule and $w$ of the fragments calculated by FGIB. We observe that the important fragments identified by FGIB with high $w$ (red and blue) indeed play crucial role for interacting with the target protein, while the fragments with low $w$ (gray) are not involved in the interactions. This analysis validates the efficacy of the proposed goal-aware fragment extraction method using FGIB and suggests the application of FGIB as a means to improve the explainability of drug discovery.

## 5    CONCLUSION

In this paper, we proposed GEAM, a fragment-based molecular generative framework for drug discovery. GEAM consists of three modules, FGIB, SAC, and GA, responsible for goal-aware fragment extraction, fragment assembly, and fragment modification, respectively. In the generative cycle of the three modules, FGIB provides goal-aware fragments to SAC, SAC provides high-quality population to GA, and GA provides novel fragments to FGIB, enabling GEAM to achieve superior optimization performance with high molecular novelty and diversity on a variety of drug discovery tasks. These results highlight its strong applicability to real-world drug discovery.

**Ethics statement** Given the effectiveness of GEAM to real-world drug discovery tasks, GEAM has the possibility to be used maliciously to generate harmful or toxic molecules. This can be prevented by setting the target properties to comprehensively consider toxicity and other side effects.

**Reproducibility statement** The code to reproduce the experimental results of this paper is available at https://anonymous.4open.science/r/GEAM-45EF. Experimental details regarding the experiments of Section 4.1 are provided in Section C.1 and Section C.2. Experimental details regarding the experiments of Section 4.2 are provided in Section C.1 and Section C.3.

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

## A GENERATION PROCESS OF GEAM

---

**Algorithm 1** A Single Generation Cycle of GEAM

---

**Input:** Fragment vocabulary $\mathcal{S}$, termination number of atoms in SAC $n_{\text{SAC}}$,
      trained FGIB, population size of GA $P$, maximum vocabulary size $L$
▷ *Fragment assembly*
Initialize $\mathbf{s}_0 = \text{benzene}$
**for** $t = 0, 1, \ldots$ **do**
  Sample $\mathbf{a}_t$ from $p_{\pi_1}, p_{\pi_2}$ and $p_{\pi_3}$ in Eq. (7-9) with $\mathcal{S}$
  Construct $\mathbf{s}_{t+1}$ by taking $\mathbf{a}_t$ on $\mathbf{s}_t$
  **if** no attachment point left in $\mathbf{s}_{t+1}$ **or** $n_{t+1} > n_{\text{SAC}}$ **then**
    $T \leftarrow t + 1$
    break
  **end if**
**end for**
Calculate the property $Y$ of $\mathbf{s}_T$
Set $r_T \leftarrow Y$
Train SAC with Eq. (10)
▷ *Fragment modification*
Initialize a population with the top-$P$ molecules generated so far
Select parent molecules from the population
Perform crossover and mutation to generate an offspring $\mathbf{o}$
Calculate the property $Y$ of $\mathbf{o}$
▷ *Fragment extraction*
Extract fragments $\mathcal{S}'$ from $\mathbf{o}$ with FGIB
Set $\mathcal{S} \leftarrow$ the top-$L$ fragments in $\mathcal{S} \cup \mathcal{S}'$ in terms of Eq. (6)
**Output:** Generated molecules ($\mathbf{s}_T$ and $\mathbf{o}$), updated vocabulary $\mathcal{S}$

---

## B PROOF OF EQUATION 5

In this section, we prove that our objective function $\mathcal{L}(\theta, \phi)$ in Eq. (5) is the upper bound of the intractable objective $\mathcal{L}_{\text{IB}}(\theta)$ in Eq. (4). At this point, we only have joint data distribution $p(G, Y)$ and the stochastic encoder $p_\theta(Z|G) = \mathcal{N}(\boldsymbol{\mu}_\theta(G), \boldsymbol{\Sigma}_\theta(G))$.

*Proof.* Following standard practice in Information Bottleneck literature (Alemi et al., 2017), we assume Markov Chains so that joint distribution $p_\theta(G, Z, Y)$ factorizes as:

$$p_\theta(G, Z, Y) = p_\theta(Z|G, Y)p(Y|G)p(G) = p_\theta(Z|G)p(Y|G)p(G). \tag{12}$$

Firstly, we derive the upper bound of the mutual information between $Z$ and $Y$:

$$I(Z, Y; \theta) = \int \int p_\theta(y, z) \log \frac{p_\theta(y, z)}{p(y)p_\theta(z)} dy dz = \int \int p_\theta(y, z) \log \frac{p_\theta(y|z)}{p(y)} dy dz,$$

where $y$ and $z$ are realization of random variables $Y$ and $Z$, respectively. The posterior is fully defined as:

$$p_\theta(y|z) = \sum_g p_\theta(g, y|z) = \sum_g p(y|g)p_\theta(g|z) = \sum_g \frac{p(y|g)p_\theta(z|g)p(g)}{p_\theta(z)},$$

where $g$ is a realization of the random variable $G$. Since this posterior $p_\theta(y|z)$ is intractable, we utilize a variational distribution $q_\phi(y|z)$ to approximate the posterior. Since KL divergence is always non-negative, we get the following inequality:

$$D_{\text{KL}}(p_\theta(Y|Z = z) \parallel q_\phi(Y|Z = z)) \geq 0 \Rightarrow \int p_\theta(y|z) \log p_\theta(y|z) dy \geq \int p_\theta(y|z) \log q_\phi(y|z) dy.$$

With this inequality, we get the lower bound of $I(Z, Y)$:

$$
\begin{aligned}
I(Z, Y; \theta) &= \int \int p_\theta(y, z) \log \frac{p_\theta(y|z)}{p(y)} dy dz \\
&\geq \int \int p_\theta(y, z) \log \frac{q_\phi(y|z)}{p(y)} dy dz \\
&= \int \int p_\theta(y, z) \log q_\phi(y|z) dy dz - \int \int p_\theta(y, z) \log p(y) dy dz \\
&= \int \int p_\theta(y, z) \log q_\phi(y|z) dy dz - \int \log p(y) \int p_\theta(y, z) dz dy \\
&= \int \int p_\theta(y, z) \log q_\phi(y|z) dy dz - \int p(y) \log p(y) dy \\
&= \int \int p_\theta(y, z) \log q_\phi(y|z) dy dz + H(Y),
\end{aligned}
$$

where $H(Y)$ is the entropy of labels $Y$. Since $Y$ is ground truth label, it is independent of our parameter $\theta$. It means the entropy is constant for our optimization problem and thus we can ignore it. By the assumption in Eq. (12),

$$
p_\theta(y, z) = \sum_g p_\theta(g, y, z) = \sum_g p(g)p(y|g)p_\theta(z|g).
$$

Thus, we get the lower bound as follows:

$$
I(Z, Y; \theta) \geq \sum_g \int \int p(g)p(y|g)p_\theta(z|g) \log q_\phi(y|z) dy dz. \tag{13}
$$

Now, we derive the upper bound of the mutual information between $Z$ and $G$:

$$
\begin{aligned}
I(Z, G; \theta) &= \sum_g \int p_\theta(g, z) \log \frac{p_\theta(z|g)}{p_\theta(z)} dz \\
&= \sum_g \int p_\theta(g, z) \log p_\theta(z|g) dz - \sum_g \int p_\theta(g, z) \log p_\theta(z) dz \\
&= \sum_g \int p_\theta(g, z) \log p_\theta(z|g) dz - \int p_\theta(z) \log p_\theta(z) dz. \tag{14}
\end{aligned}
$$

The marginal distribution $p_\theta(z)$ is intractable since

$$
p_\theta(z) = \sum_g p_\theta(z|g)p(g).
$$

We utilize another variational distribution $u(z)$ that approximate the marginal. Since KL divergence is always non-negative,

$$
D_{\mathrm{KL}}(p_\theta(Z) \| r(Z)) \geq 0 \Rightarrow \int p_\theta(z) \log p_\theta(z) dz \geq \int p_\theta(z) \log u(z) dz.
$$

Combining this inequality with Eq. (14), we get the upper bound as:

$$
\begin{aligned}
I(Z, G; \theta) &\leq \sum_g \int p_\theta(g, z) \log p_\theta(z|g) dz - \int p_\theta(z) \log u(z) dz \\
&= \sum_g \int p_\theta(g, z) \log p_\theta(z|g) dz - \int \sum_g p_\theta(z, g) \log u(z) dz \\
&= \sum_g \int p_\theta(z|g)p(g) \log \frac{p_\theta(z|g)}{u(z)}. \tag{15}
\end{aligned}
$$

Combining Eq. (13) and Eq. (15), and using the empirical data distribution $p(g,y) = \frac{1}{N}\sum_{n=1}^{N}\delta_{G_i}(g)\delta_{Y_i}(y)$, we get

$$\mathcal{L}_{\text{IB}}(\theta) = -I(Z,Y;\theta) + \beta I(Z,G;\theta)$$

$$\leq - \int\int p(g)p(y|g)p_\theta(z|g)dydz$$

$$+ \beta\sum_g \int p_\theta(z|g)p(g)\log\frac{p_\theta(z|g)}{u(z)}$$

$$\approx \frac{1}{N}\sum_{i=1}^{N}\left[\int -p_\theta(z|G_i)\log q_\phi(Y_i|z) + \beta p_\theta(z|G_i)\log\frac{p_\theta(z|g)}{u(z)}dz\right]$$

$$= \frac{1}{N}\sum_{i=1}^{N}\mathbb{E}_{p_\theta(Z|G_i)}[-\log q_\phi(Y_i|Z)] + \beta D_{\text{KL}}(p_\theta(Z|G_i)\parallel u(Z))$$

$$\approx \frac{1}{N}\sum_{i=1}^{N}(-\log q_\phi(Y_i|Z_i) + \beta D_{\text{KL}}(p_\theta(Z|G_i)\parallel u(Z)))$$

$$= \mathcal{L}(\theta,\phi),$$

where we sample $Z_i$ from $\mathcal{N}(\boldsymbol{\mu}_\theta(G_i),\boldsymbol{\Sigma}_\theta(G_i)) = p_\theta(Z|G_i)$. Therefore, we conclude that $\mathcal{L}(\theta,\phi)$ is the upper bound of $\mathcal{L}_{\text{IB}}(\theta)$. □

## C EXPERIMENTAL DETAILS

### C.1 COMMON EXPERIMENTAL DETAILS

Here, we describe the common implementation details of GEAM throughout the experiments. Following Yang et al. (2021), Lee et al. (2023b) and Gao et al. (2022), we used the ZINC250k (Irwin et al., 2012) dataset with the same train/test split used by Kusner et al. (2017) in all the experiments. To calculate novelty, we used the RDKit (Landrum et al., 2016) library to calculate similarities between Morgan fingerprints of radius 2 and 1024 bits. To calculate #Circles, we used the public code[2] and set the threshold to 0.75 as suggested by Xie et al. (2023).

**The fragment extraction module** Regarding the architecture of FGIB, we set the number of message passing in the MPNN to 3 and the number of layers of the MLP to 2. Given the perturbed fragment embedding $Z$, the property predictor $q_\phi$ first get the perturbed graph embedding with average pooling and pass it through an MLP of 3 as $\hat{Y} = \text{MLP}_\phi(\text{AvgPool}(Z))$. FGIB was trained to 10 epochs in each of the task with a learning rate of $1e-3$ and $\beta$ of $1e-5$. The initial vocabulary size was set to $K = 300$. Regarding the dynamic vocabulary update, the maximum vocabulary update in a single cycle was set to 50 and the maximum vocabulary size was set to $L = 1,000$. Following Yang et al. (2021), fragments that induce the sanitization error of the RDKit (Landrum et al., 2016) library are filtered out in the fragment extraction step.

**The fragment assembly module** Following Yang et al. (2021), we allowed GEAM to randomly generate molecules during the first 4,000 SAC steps to collect experience. Note that unlike Yang et al. (2021), we included these molecules in the final evaluation to equalize the total number of oracle calls for a fair comparison. SAC starts each of the generation episode from benzene with attachment points on the *ortho-*, *meta-*, *para-*positions, i.e., `c1(*)c(*)ccc(*)c1`. We set the termination number of atoms in the SAC to $n_{\text{SAC}} = 40$, so that an episode ends when the size of the current molecule exceeds 40. Other architectural details followed Yang et al. (2021).

**The fragment modification module** The population size of the GA was set to $P = 100$ and the mutation rate was set to 0.1. The minimum number of atoms of generated molecules was set to 15. The crossover and the mutation rules followed those of Jensen (2019).

---

[2]https://openreview.net/forum?id=Yo06F8kfMa1

## C.2 Optimization of Binding Affinity under QED, SA and Novelty Constraints

We used the RDKit (Landrum et al., 2016) library to calculate QED and SA. We used QuickVina 2 (Alhossary et al., 2015), a popular docking program, to calculate docking scores with the exhaustiveness of 1. Following Lee et al. (2023b), we first clip DS in the range $[-20, 0]$ and compute $\widehat{\text{DS}}$ and $\widehat{\text{SA}}$ to normalize each of the properties in Eq. (11) as follows:

$$\widehat{\text{DS}} = -\frac{\text{DS}}{20}, \quad \widehat{\text{SA}} = \frac{10 - \text{SA}}{9}. \tag{16}$$

In this way, each property in Eq. (11), $\widehat{\text{DS}}$, QED, $\widehat{\text{SA}}$, as well as the total property $Y$ are confined to the range $[0, 1]$.

For the baselines, we mostly followed the settings in the respective original paper. For RationaleRL, we used the official code[3]. Following the instruction, we extracted the rationales for parp1, fa7, 5ht1b, braf and jak2, respectively, then filtered them out with the QED $> 0.5$ and the SA $< 5$ constraints. Each rationale was expanded for 200 times and the model was trained for 50 iterations during the finetune. To generate 3,000 molecules, each rationale was expanded for $\lfloor \frac{3,000}{\text{\# of rationales}} \rfloor$ times, then 3,000 molecules were randomly selected. For FREED, we used the official code[4] and used the predictive error-PER model. We used the provided vocabulary of 91 fragments extracted by CReM (Polishchuk, 2020) with the ZINC250k dataset and set the target property to Eq. (11). Note that this was referred to as FREED-QS in the paper of Lee et al. (2023b). For PS-VAE, we used the official code[5] and trained the model with the target properties described in Eq. (11), then generated 3,000 molecules with the trained model. For MiCaM, we used the official code[6] to extract fragments from the ZINC250k training set. As this resulted in a vocabulary of large size and worsened the performance when applied to GEAM, we randomly selected $K = 300$ to construct the final vocabulary. The code regarding goal-directed generation is not publicly available at this time.

## C.3 Optimization of Multi-property Objectives in PMO Benchmark

We directly used the score function in each of the tasks as the property function $Y$ of FGIB. We set the number of the GA reproduction per one SAC episode to 3. For the baselines, the results in Table 5 were taken from Gao et al. (2022) and the novelty and the #Circles results in Table 6 were obtained using the official repository of Gao et al. (2022)[7].

---

[3] https://github.com/wengong-jin/multiobj-rationale
[4] https://github.com/AITRICS/FREED
[5] https://github.com/THUNLP-MT/PS-VAE
[6] https://github.com/MIRALab-USTC/AI4Sci-MiCaM
[7] https://github.com/wenhao-gao/mol_opt

Table 7: **Novel hit ratio (%) results** of additional baselines. The results are the means and the standard deviations of 3 runs. The results for the baselines are taken from Lee et al. (2023b). The best results are highlighted in bold.

| Method | Target protein | | | | |
|---|---|---|---|---|---|
| | parp1 | fa7 | 5ht1b | braf | jak2 |
| GCPN (You et al., 2018) | 0.056 (± 0.016) | 0.444 (± 0.333) | 0.444 (± 0.150) | 0.033 (± 0.027) | 0.256 (± 0.087) |
| JTVAE (Jin et al., 2018) | 0.856 (± 0.211) | 0.289 (± 0.016) | 4.656 (± 1.406) | 0.144 (± 0.068) | 0.815 (± 0.044) |
| GraphAF (Shi et al., 2019) | 0.689 (± 0.166) | 0.011 (± 0.016) | 3.178 (± 0.393) | 0.956 (± 0.319) | 0.767 (± 0.098) |
| GA+D (Nigam et al., 2020) | 0.044 (± 0.042) | 0.011 (± 0.016) | 1.544 (± 0.273) | 0.800 (± 0.864) | 0.756 (± 0.204) |
| MARS (Xie et al., 2020) | 1.178 (± 0.299) | 0.367 (± 0.072) | 6.833 (± 0.706) | 0.478 (± 0.083) | 2.178 (± 0.545) |
| GEGL (Ahn et al., 2020) | 0.789 (± 0.150) | 0.256 (± 0.083) | 3.167 (± 0.260) | 0.244 (± 0.016) | 0.933 (± 0.072) |
| GraphDF (Luo et al., 2021) | 0.044 (± 0.031) | 0.000 (± 0.000) | 0.000 (± 0.000) | 0.011 (± 0.016) | 0.011 (± 0.016) |
| LIMO (Eckmann et al., 2022) | 0.455 (± 0.057) | 0.044 (± 0.016) | 1.189 (± 0.181) | 0.278 (± 0.134) | 0.689 (± 0.319) |
| GDSS (Jo et al., 2022) | 1.933 (± 0.208) | 0.368 (± 0.103) | 4.667 (± 0.306) | 0.167 (± 0.134) | 1.167 (± 0.281) |
| GEAM (ours) | **40.567** (± 0.825) | **20.711** (± 1.873) | **38.489** (± 0.350) | **27.900** (± 1.822) | **42.950** (± 1.117) |

Table 8: **Novel top 5% docking score (kcal/mol) results** of additional baselines. The results are the means and the standard deviations of 3 runs. The results for the baselines are taken from Lee et al. (2023b). The best results are highlighted in bold.

| Method | Target protein | | | | |
|---|---|---|---|---|---|
| | parp1 | fa7 | 5ht1b | braf | jak2 |
| GCPN (You et al., 2018) | -7.464 (± 0.089) | -7.024 (± 0.629) | -7.632 (± 0.058) | -7.691 (± 0.197) | -7.533 (± 0.140) |
| JTVAE (Jin et al., 2018) | -9.482 (± 0.132) | -7.683 (± 0.048) | -9.382 (± 0.332) | -9.079 (± 0.069) | -8.885 (± 0.026) |
| GraphAF (Shi et al., 2019) | -9.327 (± 0.030) | -7.084 (± 0.025) | -9.113 (± 0.126) | -9.896 (± 0.226) | -8.267 (± 0.101) |
| GA+D (Nigam et al., 2020) | -8.365 (± 0.201) | -6.539 (± 0.297) | -8.567 (± 0.177) | -9.371 (± 0.728) | -8.610 (± 0.104) |
| MARS (Xie et al., 2020) | -9.716 (± 0.082) | -7.839 (± 0.018) | -9.804 (± 0.073) | -9.569 (± 0.078) | -9.150 (± 0.114) |
| GEGL (Ahn et al., 2020) | -9.329 (± 0.170) | -7.470 (± 0.013) | -9.086 (± 0.067) | -9.073 (± 0.047) | -8.601 (± 0.038) |
| GraphDF (Luo et al., 2021) | -6.823 (± 0.134) | -6.072 (± 0.081) | -7.090 (± 0.100) | -6.852 (± 0.318) | -6.759 (± 0.111) |
| LIMO (Eckmann et al., 2022) | -8.984 (± 0.223) | -6.764 (± 0.142) | -8.422 (± 0.063) | -9.046 (± 0.316) | -8.435 (± 0.273) |
| GDSS (Jo et al., 2022) | -9.967 (± 0.028) | -7.775 (± 0.039) | -9.459 (± 0.101) | -9.224 (± 0.068) | -8.926 (± 0.089) |
| GEAM (ours) | **-12.891** (± 0.158) | **-9.890** (± 0.116) | **-12.374** (± 0.036) | **-12.342** (± 0.095) | **-11.816** (± 0.067) |

# D ADDITIONAL EXPERIMENTAL RESULTS

## D.1 OPTIMIZATION OF BINDING AFFINITY UNDER QED, SA AND NOVELTY CONSTRAINTS

We include the novel hit ratio and the novel top 5% DS results of the additional baselines in Table 7 and Table 8. As shown in the tables, the proposed GEAM outperforms all the baselines by a large margin.

We also provide examples of the generated novel hits by GEAM for each protein target in Figure 5. The examples were collected without curation.

## D.2 OPTIMIZATION OF MULTI-PROPERTY OBJECTIVES IN PMO BENCHMARK

We provide examples of the generated top-5 molecules by GEAM for each task in Figure 6. The examples are from a single run with a random seed for each task.

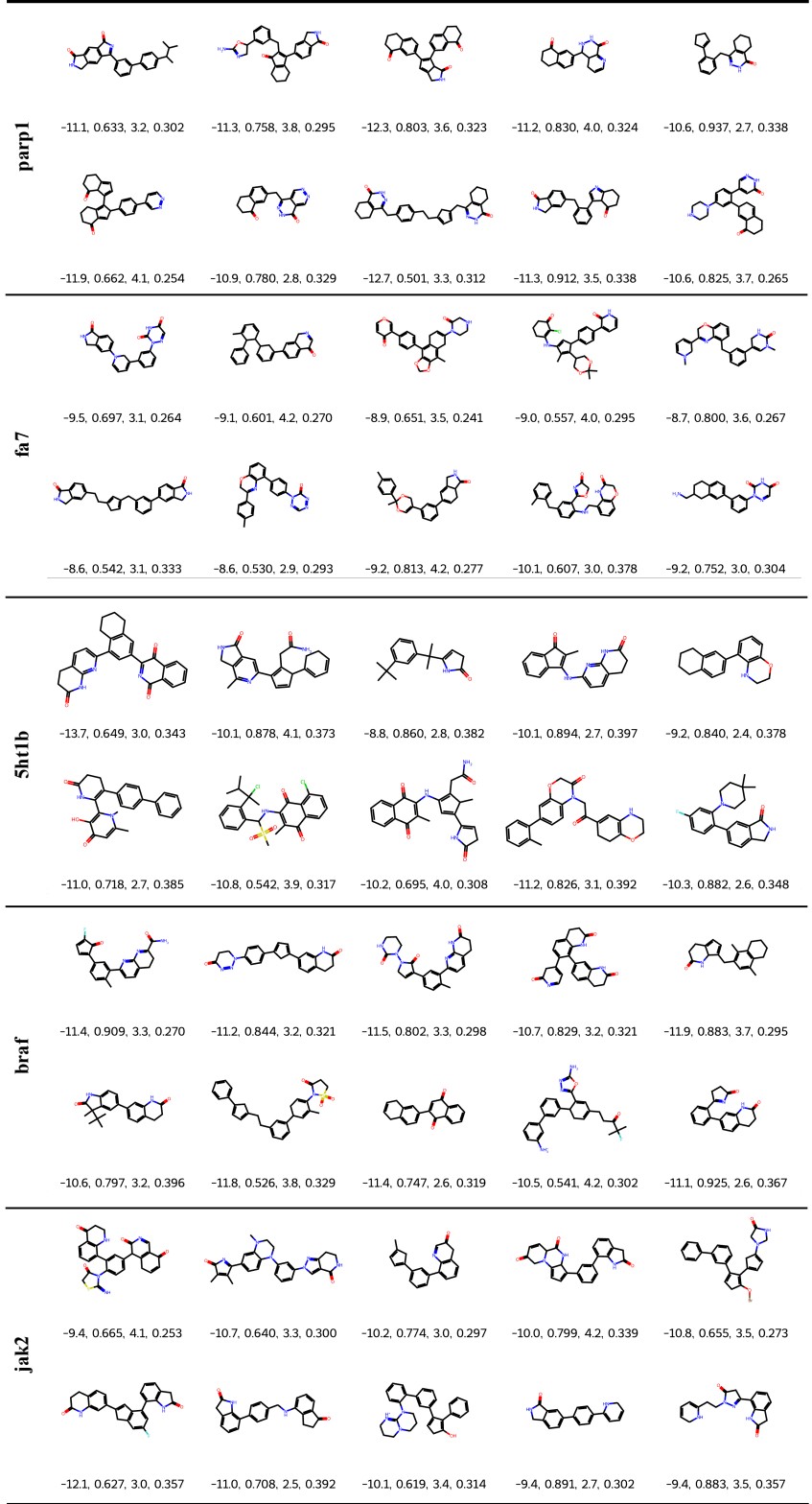

Figure 5: **The examples of the generated novel hits by GEAM.** The values of docking score (kcal/mol), QED, SA, and the maximum similarity with the training molecules are provided at the bottom of each molecule.

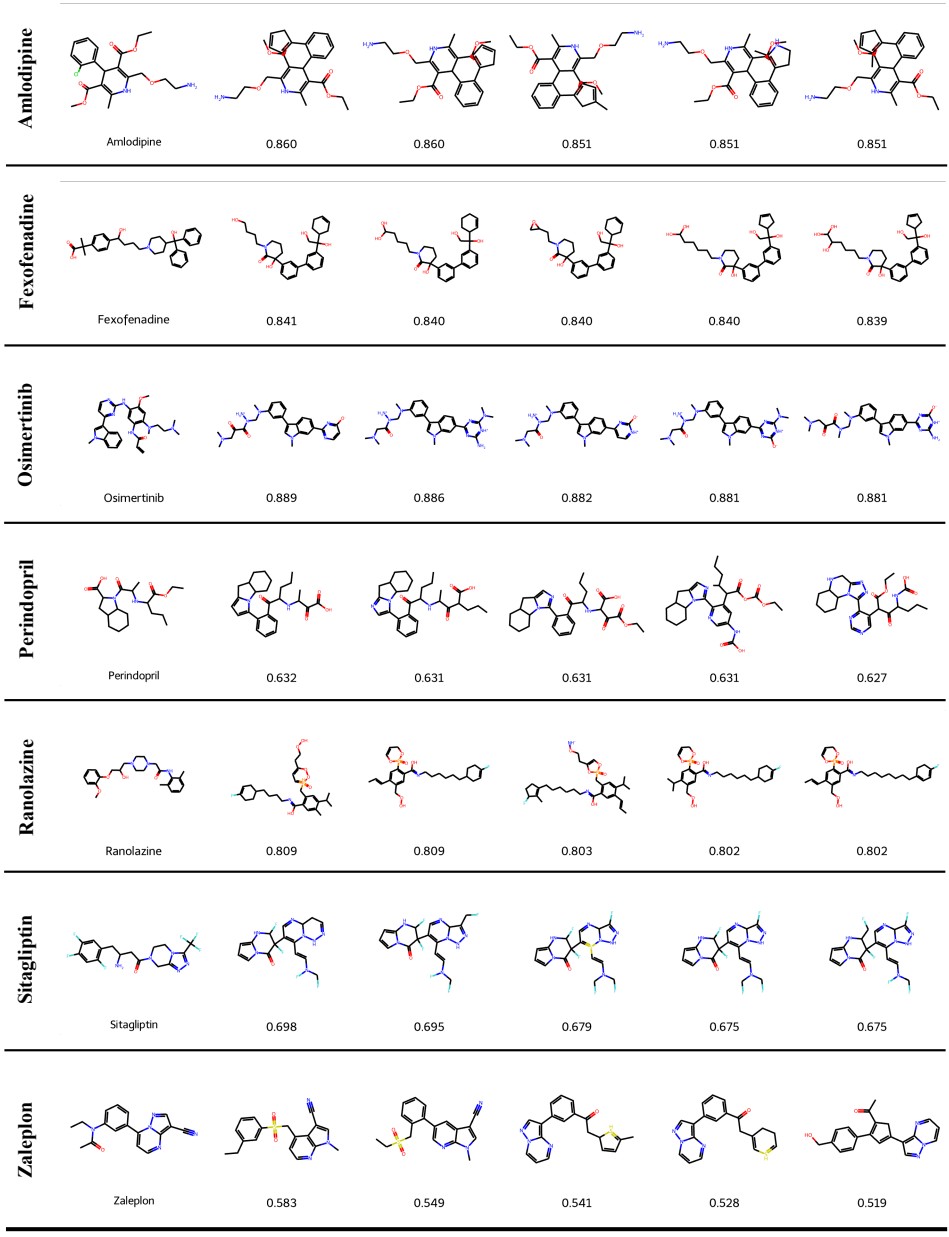

Figure 6: **The reference molecules of the PMO MPO tasks and the examples of the generated top-5 molecules from a single run of GEAM.** The scores are provided at the bottom of each generated molecule.

