# OpenReview forum: "Drug Discovery with Dynamic Goal-aware Fragments"
_ICLR.cc/2024/Conference — Submitted to ICLR 2024_

### Official Review · Reviewer_R9aj · 2023-10-31

**Soundness:** 2 fair
**Presentation:** 3 good
**Contribution:** 2 fair
**Rating:** 5
**Confidence:** 3

**Summary:**

This paper presents a molecular generation system, GEAM, that consists of a fragment generator, fragment assembler, and fragment modifier. The authors make all three components to be goal oriented. Therefore, the final system has good performance on goal-directed generation benchmarks.

**Strengths:**

- The proposed method is good motivated.
- The experiments are comprehensive.
- Overall, the paper is easy to follow.

**Weaknesses:**

The main concern is about the novelty. The three techniques, i.e., information bottleneck, soft-actor critic (SAC), and genetic algorithm (GA), are well known methods. Seems the authors just make a pipeline to combine all existing methods together.

**Questions:**

- What are the specific challenges to use information bottleneck, SAC, and GA, in molecule generation tasks? How are they mitigated?
- What are the specific challenges to use these three methods together?

---

> ### Author Response · Authors · 2023-11-16
> **Initial Response**
>
> We sincerely thank you for your comments. We appreciate your positive comments that the proposed method is well-motivated, the experiments are comprehensive, and the paper is easy to follow. We address your concerns below.
>
> ---
>
> **Comment**
>
> The main concern is about the novelty. The three techniques, i.e., information bottleneck, soft-actor critic (SAC), and genetic algorithm (GA), are well known methods. Seems the authors just make a pipeline to combine all existing methods together. What are the specific challenges to use information bottleneck, SAC, and GA, in molecule generation tasks? How are they mitigated? What are the specific challenges to use these three methods together?
>
> **Response**
>
> We want to emphasize that the contribution of our work lies in the **identification of the specific limitations in existing fragment-based drug discovery (FBDD) models and combination of the fragment extraction, fragment assembly and fragment modification modules in a novel way to overcome the limitations.** Specifically, we have pointed out that existing FBDD models have the following limitations: they do not take the target chemical properties into account when extracting fragments, rely on heuristic fragment selection rules, and/or the exploration is confined to the initial fragment vocabulary.
>
> To effectively overcome these, we have proposed the combination of the three modules that were carefully chosen for how they combine: the fragment extraction module provides goal-aware fragments to the fragment assembly module, the fragment assembly module provides high-quality population to the fragment modification module, and the fragment modification module provides novel fragments to the fragment extraction module. Through the extensive ablation studies, we have shown that the outstanding performance of GEAM **cannot be achieved by each individual component of GEAM alone**, and that **only the proposed combination of the three modules can effectively explore the chemical space to discover novel and diverse drug candidates.**
>
> Moreover, even at the component level, we have newly proposed **FGIB as the fragment extraction module, which considers fragments as its substructure recognition unit**. This feature makes FGIB more suitable for constructing fragment vocabularies for subsequent generation, in contrast to previous GIB-based substructure recognition methods that commonly recognize disconnected substructures in a single molecule. Additionally, we have  proposed to construct the initial population of GA with molecules generated by SAC, which significantly improves exploration in chemical space compared to Graph GA [1] that draws its initial population from the training set.
>
> We believe that combining existing methods to establish a new framework and showing its emergent effects is a significant research contribution, rather than a trivial one. While Stable Diffusion [2] can be thought of as nothing more than combining the already well-known VQ-VAE [3,4] and diffusion [5], its demonstrated practical impact highlights the significance of its research contribution. Since we proposed to integrate the three modules into a single framework and experimentally validated its effectiveness, we believe that our work is making an important contribution that can have a practical impact and spawn many interesting future studies.
>
> ---
>
> **References**
>
> [1] Jensen, A graph-based genetic algorithm and generative model/monte carlo tree search for the exploration of chemical space, Chemical science, 2019.
>
> [2] Rombach et al., High-resolution image synthesis with latent diffusion models, CVPR, 2022.
>
> [3] Oord et al., Neural discrete representation learning, NeurIPS, 2017.
>
> [4] Razavi et al., Generating diverse high-fidelity images with vq-vae-2, NeurIPS, 2019.
>
> [5] Sohl-Dickstein et al., Deep unsupervised learning using nonequilibrium thermodynamics, ICML, 2015.

---

### Official Review · Reviewer_V1Zp · 2023-10-31

**Soundness:** 3 good
**Presentation:** 3 good
**Contribution:** 3 good
**Rating:** 8
**Confidence:** 3

**Summary:**

The authors proposed GEAM, a molecular generative framework for drug discovery inspired by the fragment-based drug discovery (FBDD) strategy used to explore the vast chemical space. GEAM distinguishes itself from other computational FBDD approaches by its unique approach to selecting fragments. It employs a Fragment-wise Graph Information Bottleneck (FGIB) module, drawing from the principles of graph information bottleneck (GIB). This module identifies fragments that are particularly relevant to the target chemical property. The extracted fragments are exploited by a generative model consisting of a fragment assembly module that generates new molecules from the fragments and a fragment modification module that outputs new molecules with new fragments that were not present in the original fragment vocabulary. These new fragments can then be assessed by the fragment extraction module to expand GEAM's exploration of the chemical space. The reported experiments seem to show that GEAM has better performance at discovering new drug candidates than existing molecular optimization methods, ablation studies also show the importance of each module of GEAM for effective molecular generation.

**Strengths:**

- The proposed Fragment extraction module stands out as a robust strategy for constructing the fragment vocabulary. It ensures that the generative model prioritizes the most relevant fragments for the target property. Additionally, leveraging molecules generated by the modification module to expand the fragment vocabulary further bolsters this strategy.
- The paper is well written and gives a good intuition of each module of GEAM, justifying their choice and their roles within the generative process.
- The paper's treatment of related works is comprehensive. It firmly situates the research in the context of prior work on drug discovery.
- Strong experimental evidence of the efficacy of the model for molecular generation, with ablation studies that highlight the importance of each module.
- The experiment section shows the importance of a proper selection of fragments for molecule generation, with representations learned through FGIB proving highly suitable for this task.
- Emphasis on the vital role of goal-driven fragment selection and the value of generating molecules as assemblies of those fragments, rather than solely focusing on fragment modification.
- Generative algorithms chosen for the modification module not only help to generate novel molecules but also feed FGIB to extract new instances for the fragment vocabulary.

**Weaknesses:**

- While the paper introduces GEAM as an innovative framework, where the importance of each module is well understood, it could benefit from a more explicit delineation of the unique contributions of each module, mainly for the assembly and modification modules.
- The paper could enhance its clarity by describing the specific advantages of FGIB over existing substructure identification architectures based on GIB theory.
- Lack of clear intuition for the loss proposed in Equation 5, focusing on how the optimization of the first term contributes to the identification of the important fragments.

It's important to note that while these weaknesses have been identified, I do not think they are a big issue for the acceptance of the paper.

**Questions:**

See weaknesses.
- The set of properties Y for each molecule in the training set is generated using Equation 11. Could you please provide more information on how the terms QED and SA are computed for each molecule?
- Some papers in the bibliography are cited with the arXiv version while there exists a peer-reviewed version. Is this because the arXiv version contains additional information not present in the peer-reviewed version, or is there another reason?

**Details Of Ethics Concerns:**

The authors recognize the potential of GEAM in the generation of harmful or toxic molecules in their ethics statement (Potentially harmful applications).

---

> ### Author Response · Authors · 2023-11-16
> **Initial Response (1/2)**
>
> We sincerely appreciate your positive comments that our method distinguishes itself from existing methods by its unique approach to selecting fragments, the paper is well-written and gives a good intuition, the related works are comprehensive, and the experimental results are strong. We address your concerns below. We have also revised our paper to reflect your comments, and highlighted the modified parts in $\color{blue}{\text{blue}}$.
>
> ---
>
> **Comment 1**
>
> While the paper introduces GEAM as an innovative framework, where the importance of each module is well understood, it could benefit from a more explicit delineation of the unique contributions of each module, mainly for the assembly and modification modules.
>
> **Response 1**
>
> We emphasize that the contribution of our work lies in the **identification of the specific limitations in existing fragment-based drug discovery (FBDD) models and combination of the fragment extraction, fragment assembly and fragment modification modules in a novel way to overcome the limitations.** Specifically, we have pointed out that existing FBDD models have the following limitations: they do not take the target chemical properties into account when extracting fragments, rely on heuristic fragment selection rules, and/or the exploration is confined to the initial fragment vocabulary.
>
> To effectively overcome these, we have proposed the combination of the three modules that were carefully chosen for how they combine: the fragment extraction module provides goal-aware fragments to the fragment assembly module, the fragment assembly module provides high-quality population to the fragment modification module, and the fragment modification module provides novel fragments to the fragment extraction module. Through the extensive ablation studies, we have shown that the outstanding performance of GEAM cannot be achieved by each individual component of GEAM alone, and that **only the proposed combination of the three modules can effectively explore the chemical space to discover novel and diverse drug candidates.**
>
> ---
>
> **Comment 2**
>
> The paper could enhance its clarity by describing the specific advantages of FGIB over existing substructure identification architectures based on GIB theory.
>
> **Response 2**
>
> The advantage of the proposed FGIB is that **its substructure recognition unit is fragments rather than atoms, which makes FGIB more suitable for constructing fragment vocabularies for subsequent generation.** As we explained in the paper, subgraph recognition by GIB has been only employed in classification and regression tasks, and this is the first work that applies GIB to fragment extraction. In practice, previous subgraph recognition methods are inadequate for extracting connected fragments, as they commonly recognize disconnected or atomic substructures in a single molecule. We have additionally included this explanation in Related Work.
>
> ---
>
> **Comment 3**
>
> Lack of clear intuition for the loss proposed in Equation 5, focusing on how the optimization of the first term contributes to the identification of the important fragments.
>
> **Response 3**
>
> **Optimization of both the first and second terms in Equation 5 contributes to the identification of the important fragment.** The first term and the second term are the upper bounds of $-I(Z,Y;\theta)$ and $\beta I(Z,G;\theta)$, respectively. Intuitively, minimizing $-I(Z,Y;\theta)$ encourages representation of chosen fragments to be expressive for predicting a label $Y$ while minimizing $\beta I(Z,G;\theta)$ encourages choosing sparse fragments to forget a graph $G$. Therefore, optimizing only the first terms without considering the second term, which enforces the selection of sparse fragments, leads to a trivial solution where the model chooses all the fragments of the graph to predict the label.
>
> ---
>
> **Comment 4**
>
> The set of properties Y for each molecule in the training set is generated using Equation 11. Could you please provide more information on how the terms QED and SA are computed for each molecule?
>
> **Response 4**
>
> Following many existing works and all the baselines of our paper, QED and SA were computed using the RDKit library [1]. We have additionally included this detail in Section C.2.
>
> ---
>
> **Comment 5**
>
> Some papers in the bibliography are cited with the arXiv version while there exists a peer-reviewed version. Is this because the arXiv version contains additional information not present in the peer-reviewed version, or is there another reason?
>
> **Response 5**
>
> We have corrected the citation of the Yang et al. paper [2]. Thank you for pointing this out.

---

> ### Author Response · Authors · 2023-11-16
> **Initial Response (2/2)**
>
> **Ethics Concerns**
>
> The authors recognize the potential of GEAM in the generation of harmful or toxic molecules in their ethics statement (Potentially harmful applications).
>
> **Response 6**
>
> The risk of generating toxic samples is a common problem shared by all generative models across domains such as molecules, graphs, images and languages. However, this can be effectively prevented by filtering out toxic fragments from the generation cycle or by incorporating an auxiliary toxicity detection module that filters out the generated toxic molecules in the model service stage — similar to how ChatGPT filters out hateful or discriminatory sentences (please refer to this [link](https://platform.openai.com/docs/guides/moderation)).
>
> ---
>
> **References**
>
> [1] Landrum et al., RDKit: Open-source cheminformatics software, 2016. *URL https://www.rdkit.org*
>
> [2] Yang et al., Hit and lead discovery with explorative rl and fragment-based molecule generation, NeurIPS, 2021.

---

> ### Comment · Reviewer_V1Zp · 2023-11-22
>
> I would like to thank the authors for the clarifications and modifications, all my concerns were properly addressed.

---

### Official Review · Reviewer_bdP8 · 2023-11-01

**Soundness:** 3 good
**Presentation:** 3 good
**Contribution:** 3 good
**Rating:** 6
**Confidence:** 4

**Summary:**

The authors designed a molecular generation framework for drug discovery called GEAM. GEAM consists of three modules, which are responsible for goal-aware fragment extraction, fragment assembly, and fragment modification.GEAM achieves leading results on drug discovery tasks.

**Strengths:**

- The research content of the article is drug discovery, which is an important and cutting-edge field.

- The method proposed in the article is simple and effective. The theoretical analysis and proof are clear.

- The article provides detailed experimental results.

**Weaknesses:**

In the experimental part, the method proposed in the article did not achieve the best results on some data or some indicators.

**Questions:**

The method will add the fragments generated by the model to the vocabulary.

Will this introduce illusion or false information?

Should there be a section to verify the reliability and authenticity of the generated snippets? This may be the reason why experimental results do not show consistent superiority.

**Details Of Ethics Concerns:**

Preventing the generation of toxic drugs by setting target attributes is a subjective behavior. If a user maliciously actively generates toxic drugs, how should it be prevented? During model training, are there special treatments for fragments that may produce toxic side effects?

---

> ### Author Response · Authors · 2023-11-16
> **Initial Response**
>
> We sincerely thank you for your comments. We appreciate your positive comments that the proposed method is effective, the clarity of theoretical analysis and proofs and the detailed experimental results. We address your concerns below.
>
> ---
>
> **Comment 1**
>
> In the experimental part, the method proposed in the article did not achieve the best results on some data or some indicators.
>
> **Response 1**
>
> In our paper, **we evaluated the effectiveness of our proposed GEAM on a total of twelve tasks and showed that GEAM performed best on eleven tasks**. In the five binding affinity optimization tasks, GEAM outperformed all the baselines by a very large margin (Table 1 and Table 2). In the PMO MPO tasks, GEAM achieved the best results in six out of the seven tasks (Table 5). In addition, the novelty and diversity values of GEAM are at the best level compared to the baselines except MORLD and REINVENT  (Table 3, Table 4 and Table 6). As we explained in the paper, the high novelty values of MORLD are trivial due to its poor optimization performance and very low diversity, and the high diversity values of RationaleRL on the target proteins 5ht1b and jak2 are not meaningful due to its poor optimization performance and novelty.
>
> We emphasize that as we explained in the paper, there is a general trend that the more powerful the molecular optimization model, the less likely it is to generate diverse molecules [1], but our proposed GEAM overcomes this trade-off by discovering novel and high-quality goal-aware fragments on-the-fly. Among the models tested in our paper, **GEAM is the only one that consistently showed excellent performance in all three aspects of optimization performance, novelty and diversity, suggesting that GEAM is highly suitable for drug discovery.**
>
> ---
>
> **Comment 2**
>
> The method will add the fragments generated by the model to the vocabulary. Will this introduce illusion or false information? Should there be a section to verify the reliability and authenticity of the generated snippets?
>
> **Response 2**
>
> We have qualitatively and quantitatively examined the effectiveness of generated fragments. Through the qualitative analysis in Figure 3(d), **we have shown that the fragments extracted by the FGIB module match well with chemical knowledge**. We also have performed extensive ablation studies to verify the quality of the extracted fragments by the FGIB module (Figure 3(a) and Figure 3(b)). In particular, when comparing GEAM and GEAM-static, we can observe that adding fragments extracted during generation does not degrade performance, but improves novelty and diversity. Overall, we can conclude that **the FGIB module and the extracted fragments by the module are reliable and help generate novel and diverse drug candidates.**
>
> ---
>
> **Ethics Concerns**
>
> Preventing the generation of toxic drugs by setting target attributes is a subjective behavior. If a user maliciously actively generates toxic drugs, how should it be prevented? During model training, are there special treatments for fragments that may produce toxic side effects?
>
> **Response 3**
>
> The risk of generating toxic samples is a common problem shared by all generative models across domains such as molecules, graphs, images and languages. However, this can be effectively prevented by filtering out toxic fragments from the generation cycle or by incorporating an auxiliary toxicity detection module that filters out the generated toxic molecules in the model service stage — similar to how ChatGPT filters out hateful or discriminatory sentences (please refer to this [link](https://platform.openai.com/docs/guides/moderation)).
>
> ---
>
> **References**
>
> [1] Gao et al., Sample efficiency matters: a benchmark for practical molecular optimization, NeurIPS, 2022.

---

### Author Response · Authors · 2023-11-21
**A Gentle Reminder**

Dear reviewers,

We appreciate your positive comments that our work is effective and the theoretical analysis is clear (Reviewer bdP8), that the paper is well-written and gives a good intuition of each module through strong experimental evidence (Reviewer V1Zp), and that the method is well-motivated and the experiments are comprehensive (Reviewer R9aj).

In addition, we sincerely appreciate your comments to strengthen our work. We have made every effort to faithfully address all your comments in the responses. As we approach the end of the rebuttal period, please review our responses and let us know if there is anything else we should address further.

Best regards, Authors

---

### Meta-Review · Area_Chair_ucAA · 2023-12-06

**Metareview:**

This paper introduces GEAM, a molecular generation framework for drug discovery. GEAM is a molecular generation framework that integrates goal-aware fragment extraction, assembly, and modification procedures to enhance drug discovery. The Fragment-wise Graph Information Bottleneck (FGIB) module is a standout feature of the proposed approach, effectively selecting fragments relevant to desired chemical properties. Notably, the paper offers robust experimental evidence supporting the superior performance of GEAM in generating novel and diverse drug candidates, compared to pre-existing methods.

**Strengths:**
- The paper has been commended for effectively addressing the current limitations of fragment-based drug discovery (FBDD) models with coherent integration of three modules: fragment extraction, assembly, and modification.
- This paper provides good experimental design, the clear and insightful presentation style, and the comprehensive treatment of related work.

**Weaknesses:**
-  2 of 3 reviewers raised concerns about the novelty given that the methods employed are not new individually. I regard this argument as valid and I was not convinced by the response from the reviewers. " Specifically, we have pointed out that existing FBDD models have the following limitations: they do not take the target chemical properties into account when extracting fragments," is not true because RationalRL is an RL model that first identifies subgraphs that are likely responsible for the target properties.
- Missing of related work, several important related works are missing. For example, the retrieve-based drug design[1] and the structural-based molecular optimization[2] should be included and compared in your experimental part. I think the retrieval-based mol gen is quite related to the proposed model intuitively.
- The benchmark for comparison is not quite widely used. I noticed the previous work of MOOD but I think it is better if you choose some widely-used benchmarks in multi-objective molecular generation.

[1] RETRIEVAL-BASED CONTROLLABLE MOLECULE GENERATION. ICLR 23.
[2] Reinforced Genetic Algorithm for Structure-based Drug Design. NeurIPS 22.

**Justification For Why Not Higher Score:**

Comparisons with more related works mentioned above and reporting results on widely used multi-objective molecular generation benchmarks will make this manuscript more strong.

**Justification For Why Not Lower Score:**

N/A

---

### Decision · Program_Chairs · 2024-01-16

Reject